# Clinical, Cytogenetic and Molecular Cytogenetic Outcomes of Cell-Free DNA Testing for Rare Chromosomal Anomalies

**DOI:** 10.3390/genes13122389

**Published:** 2022-12-16

**Authors:** Seher Basaran, Recep Has, Ibrahim Halil Kalelioglu, Tugba Sarac Sivrikoz, Birsen Karaman, Melike Kirgiz, Tahir Dehgan, Tugba Kalayci, Bilge Ozsait Selcuk, Peter Miny, Atil Yuksel

**Affiliations:** 1Department of Medical Genetics, Istanbul Faculty of Medicine, Istanbul University, Istanbul 34093, Turkey; 2Center for Genetic Diagnosis and Research PREMED, Istanbul 34394, Turkey; 3Department of Obstetrics and Gynecology, Istanbul Faculty of Medicine, Istanbul University, Istanbul 34093, Turkey; 4Child Health Institute, Department of Pediatric Basic Sciences, Istanbul University, Istanbul 34093, Turkey; 5Medical Genetics, Institute of Medical Genetics and Pathology, University Hospital Basel, CH-4031 Basel, Switzerland

**Keywords:** cell-free DNA, NIPT, rare chromosomal anomalies, mosaicism, positive predictive values

## Abstract

The scope of cell-free DNA (cfDNA) testing was expanded to the genome, which allowed screening for rare chromosome anomalies (RCAs). Since the efficiency of the test for RCAs remains below the common aneuploidies, there is a debate on the usage of expanded tests. This study focuses on the confirmatory and follow-up data of cases with positive cfDNA testing for RCAs and cases with screen-negative results in a series of 912 consecutive cases that underwent invasive testing following cfDNA testing. Chorion villus sampling (CVS), amniocentesis (AS), fetal blood sampling, and term placenta samples were investigated using classical cytogenetic and molecular cytogenetic techniques. Out of 593 screen-positive results, 504 (85%) were for common aneuploidies, 40 (6.7%) for rare autosomal trisomies (RATs), and 49 (8.3%) for structural chromosome anomalies (SAs). Of the screen-positives for RATs, 20 cases were evaluated only in fetal tissue, and confined placental mosaicism (CPM) could not be excluded. Among cases with definitive results (*n* = 20), the rates of true positives, placental mosaics, and false positives were 35%, 45%, and 10%, respectively. Among screen-positives for SAs, 32.7% were true positives. The confirmation rate was higher for duplications than deletions (58.3% vs. 29.4%). The rate of chromosomal abnormality was 10.9% in the group of 256 screen-negatives with pathological ultrasound findings. This study provides further data to assess the efficiency of expanded cfDNA testing for RATs and SAs. The test efficiency for cfDNA seems to be higher for duplications than for deletions, which is evidence of the role of expert ultrasound in identifying pregnancies at increased risk for chromosome anomalies, even in pregnancies with screen-negatives. Furthermore, we discussed the efficiency of CVS vs. AC in screen-positives for RATs.

## 1. Introduction

The implementation of cell-free DNA (cfDNA) testing changed the algorithms in prenatal screening for chromosome anomalies. It replaced biochemical maternal serum screening tests to become the first-tier screening test for common aneuploidies (chromosomes 21, 18, 13, X, and Y) in some countries with national regulations (e.g., in the Netherlands, Belgium) or guidelines of international professional organizations [1,2,3,4,5].

The two main technical approaches, assessing the amount of chromosome-specific DNA reads or genotyping single nucleotide polymorphisms (SNPs), require massively parallel sequencing, which can be targeted or genome-wide. Testing of cfDNA was initially validated for screening common aneuploidies [6,7] and welcomed because of the high detection rates. Genome-wide approaches allowed screening for rare autosomal trisomies (RATs), structural chromosomal anomalies (SAs) greater than 7–10 Mb, and some known microdeletion/duplication syndromes (MMSs) [8,9,10]. Rare chromosome anomalies (RCAs) constitute 16% of all fetal chromosome anomalies [11]. Limited studies with expanded tests revealed lower positive predictive values (PPV) for RCAs than for common aneuploidies [12] since the lower prevalence of RCAs decreases the PPV. Despite the consensus on the need to test recommendations for common aneuploidies, there is debate regarding expanded testing because false positives (FPs) would increase the rates of invasive procedures in clinical practice [12,13].

Almost all non-mosaic autosomal aneuploidies (except trisomy (T) 21) are nonviable, and only mosaic forms may survive. In the EUROCAT registry, of the 141 mosaic trisomies (0.60/10,000 births), 78% were detected prenatally, and 41% were live-born [11]. Mosaicism occurring due to the “trisomy rescue” mechanism is more frequently associated with RATs. Mosaicism in fetal tissues (true fetal mosaicism; TFM) may impair physical and intellectual development depending on the tissue involved, the proportion of cells affected, and the involved chromosome [14]. Mosaicism detected in chorionic villi (CV) but not in fetal tissues is termed “confined placental mosaicism (CPM),” a well-known biological phenomenon since the introduction of CV sampling (CVS) [15,16,17]. Three different types of CPM are defined; CPM-1 (abnormality only in cytotrophoblasts); CPM-2 (abnormality only in the mesenchyme); and CPM-3 (abnormality in both cytotrophoblasts and mesenchyme) [18]. Mosaicism is observed in about 1 to 2% of CVSs. Of those, 72 to 87% are CPM. However, the likelihood of TFM is still about 13% in CPM cases [19,20]. CPM-2 and -3 are more likely to be associated with TFM, and therefore differential diagnosis between the subgroups of mosaicism is important. Interphase fluorescence in-situ hybridization technique (I-FISH) with relevant probes is helpful in searching the mosaicism [21]. There is still a debate about whether or not CPM impairs placental functions and affects fetal growth [22,23,24]. Benn et al. [25] reviewed ten reports with positive cfDNA testing for RATs. Among cases with follow-up evaluation, 41.1% resulted in a normal live birth, and the fetal loss rate was 27.2%. Abnormal fetal ultrasound findings or phenotypical abnormalities were observed in 7.3% of the newborns. Fetal growth restriction (FGR) or low birth weight and uniparental disomy (UPD) were detected in 14.6% and 2.0%, respectively. If expanded cfDNA testing is positive for RAT, the invasive diagnostic test in clinical practice is amniocentesis (AC) to avoid the CPM of CVS [26]. Confirmation of all the screen-positives and follow-up studies, including term placenta and examination of the fetus/newborn after the termination/delivery, would help explain the false positive or false negative cfDNA screening tests [27].

Di Renzo et al. [12] pointed out that currently, available microdeletion panels do not include all clinically significant copy number variations (CNVs). On the other side, some authors argue that there are no other screening tests for MMSs, and ultrasound, a part of standard obstetric care, may not usually be efficient in early detection [28]. Srebniak et al. [29] observed a decrease in the prenatal detection of submicroscopic aberrations, parallel to the decrease in invasive tests after the introduction of cfDNA testing as the first-tier test for common aneuploidies.

Here, we present confirmatory test results, fetal ultrasound findings, and clinical follow-up of the screen-positives for RCAs from the perspective of a tertiary care center. We discuss the efficiency of the different invasive procedures and laboratory techniques in screen-positives for RATs and SAs. Furthermore, we share the cytogenetic findings of the screen-negative cases for common aneuploidies evaluated due to the ultrasound pathologies. Since the biological nature is complex and the frequency of the RCAs is unknown in early ongoing pregnancies, sharing the experiences with positive cfDNA testing for RCAs might be helpful to improve the diagnostic approaches and to give adequate pre- and post-test genetic counseling.

## 2. Materials and Methods

This retrospective observational study includes detailed outcome data of positive cfDNA testing results for RCAs and negatives for common aneuploidies. Cases are part of a series with 912 consecutive patients investigated between November 2013 and November 2022, and 101 cases have been published previously [30]. Tests of cfDNA were offered by obstetricians primarily working in outer private practice/hospitals and were performed by internationally approved commercial providers, including BGI, Life Codexx-Sequenom, Natera, Ariosa, Illumina, Genoma, MGI, NIPD Genetics, Eurofins. Pretest counseling was given by the obstetrician, who ordered the cfDNA testing. The families were referred for confirmatory studies following a positive cfDNA test result for RCAs. Cases with US pathologies but negative screening test results for common aneuploidies were also included in the study. After a detailed US examination by experienced specialized operators, all families were counseled by a perinatologist or a medical geneticist about the confirmatory testing options and follow-up studies. All patients consented to an invasive procedure, i.e., CVS, AC, or fetal blood sampling (FBS).

Laboratory work and post-test genetic counseling were performed at the PREMED Genetic Diagnosis Center and the Department of Medical Genetics of Istanbul Medical Faculty, Istanbul University. CVS samples were investigated simultaneously by direct preparation (DP) and long-term cell culture (CC). AC and FBS samples were analyzed by CC. According to the availability of relevant probes for RATs, interphase fluorescence hybridization (I-FISH) was applied on fresh amniocytes and DP-CV slides, and at least 80 nuclei were scored. Fetal blood samples were cultured for 2–3 days, and at least 30 cells were analyzed/counted. If no anomaly was found, more than 20 additional metaphases were counted in cases with a risk for RAT. The three-five samples from different regions of the term placentas were investigated by cell culture or I-FISH technique using relevant probes.

The results of cfDNA testing were classified as “true positive (TP)”, “false positive (FP)”, and “false negative (FN)”. If the predicted anomaly was present in 2 metaphases from at least two culture flasks of AC, it was considered as “true fetal mosaic (TFM)” and TP. If the predicted anomaly was present in CVS but not shown in AC, this was classified as “CPM.” Cases without CVS or term placenta investigation were classified as FP/CPM because CPM could not be excluded. Molecular karyotyping was offered to patients with a risk for SAs, and either the Affymetrix CytoScan 315K Array (Affymetrix, Santa Clara, CA) or Agilent SurePrint G3 CGH + SNP Microarray Kit (180K) (Agilent Technologies Inc., Santa Clara, CA, USA) were used. Aberration calls were applied according to the manufacturers’ recommendations as thresholds for copy numbers; 100 kb gains and 50 kb losses across the genome. The log2 ratio was 0.5, and the minimum probe binding was 5. Uniparental disomy (UPD) was excluded by SNP genotyping with in-house designed primers (Miseq-Illumina) (Intergen, Ankara, Turkiye). For the follow-up studies, the obstetrician or families were contacted by phone. If possible, babies were examined clinically in our polyclinic.

The cfDNA tests were not covered by the governmental health system during the study period. The study was approved by the local Ethics Committee of the Istanbul Faculty of Medicine (No: 52410, Date: 28 January 2021).

## 3. Results

Out of the 912 consecutive cases that underwent invasive testing following cfDNA testing, 593 (65%) were screen-positives. Among screen positives, 504 (85%) were for common aneuploidies, 40 for RATs (6.8%), and 49 for SAs (8.3%). Two hundred fifty-six screen-negatives for common aneuploidies (28.1%) were evaluated cytogenetically due to the pathological ultrasound findings. The screening test was inconclusive or failed in the remaining 63 cases.

### 3.1. Rare Autosomal Trisomies

In the group of RATs, the mean maternal age (MA) was 34.6 (range 25–43), and the mean gestational age (GA) was 17.1 weeks (range 12–22). The most frequently predicted aneuploidies were T7 (*n* = 8) and T16 (*n* = 7), which were followed by T22 (*n* = 6), T3 (*n* = 4), T10 and T15 (*n* = 3), T4, T8 and T20 (*n* = 2), and single cases for T9, T12 and monosomy 7. All demographic, ultrasound, cytogenetic, and clinical follow-up data of the pregnancies were given in Appendix A and summarized in Table 1. CVS was applied in 13 cases, and in nine of them, a supplementary AC was performed. Predicted trisomies were confirmed in four cases (TPs). Two TP cases (T10 and T12) having abnormal ultrasound findings (Appendix A) were investigated only by CVS. Two TP cases (T16 and T22) with fetal growth restriction (FGR) or minor ultrasound findings (single umbilical artery, echogenic intracardiac focus) were investigated by a supplemental AC, which confirmed true mosaicism. These four pregnancies were terminated. Three predicted anomalies (T4, T7, T22) were not confirmed in CVS (FP). These pregnancies were delivered at term, and the newborns were healthy.

The predicted anomalies (2xT16, 2x T20, T10, T22) in six pregnancies were confirmed in CVS (direct preparation and cell culture) while not confirmed in AC. These cases were defined as CPM. Among the CPMs, one was CPM-1, and the others were CPM-3. FGR was observed in two of CPM-3 cases. One pregnancy with predicted T16 was terminated early due to the severe FGR, and postmortem physical examination revealed dysmorphic findings compatible with mosaic T16. One uneventful pregnancy with CPM-3 for T16 ended in a healthy child. FGR was observed at 31st GA in CPM-3 for T22, and a phenotypically normal baby was born at 36th GA. Two T20 pregnancies with CPM-3 are still ongoing.

Predicted trisomies were confirmed in three of twenty-seven cases investigated initially by AC. In one case (T15) having severe malformations, the cytogenetic investigation revealed an unbalanced product of a paternal translocation which was leading to a duplication of 15q21->qter (about 53 Mb) and a microdeletion of 2q37.3->qter (less than one Mb), and this pregnancy was terminated. The microdeletion was not detected/reported by cfDNA testing. Two cases with normal ultrasound findings were low-level true mosaics (T3 and T7), and UPD7 was excluded. Families opted to continue the pregnancies, and phenotypically normal babies were born. Term placenta samples could be evaluated in four cases with normal AC results. Predicted anomalies were confirmed in three of them (T4, T7, T9). These cases were reclassified as CPM. The pregnancy with predicted T7 was terminated due to abnormal ultrasound findings, including FGR. The findings in postmortem physical examination were compatible with Silver-Russel syndrome. Cases with CPM for T4 and T9 had normal ultrasound findings on the day of AC. However, FGR was observed at 32nd GA in the case of T4; the phenotypically normal baby was delivered at 36th GA. The pregnancy with CPM T9 was uneventful, and a phenotypically normal baby was born at term. A placental study revealed normal results in one case with predicted T3, and this case was classified as FP. In the group with definitive cytogenetic results (*n* = 20), the rates of TPs, CPMs, and FPs were 35%, 45%, and 10%, respectively.

The possibility of CPM could not be excluded in the remaining 20 cases classified as FP/CPM (Table 1). Ten pregnancies resulted in healthy newborns following uneventful pregnancies, but placental samples were unavailable. Seven pregnancies are ongoing and placental studies are planned. One pregnancy ended with early delivery at 33rd GA, and the baby had dermatological problems and hypospadias. One pregnancy ended with spontaneous abortion seven days after AC, and one pregnancy with early delivery at 24th GA; unfortunately, the newborn died three days later.

In the group of RATs evaluated initially by AC, the cfDNA results could be confirmed in only three cases (the rate of TP 11.1%). In the remaining cases, the possibility of CPM was not excluded.

### 3.2. Structural Chromosome Anomalies (SAs)

In the group of SAs, the mean MA was 35.4 (range 27–42), and the mean GA was 15.9 weeks (range 12–25) on the day of the biopsy. Of 49 cases with predicted SCA in cfDNA testing, 16 were confirmed (TP) (PPV 32.7% per case). All data were shown in Appendix A, and diagnostic findings were summarized in Table 2. In two cases, predicted duplications (Cases 11, 16) were unbalanced products of maternal translocations, and two deletions (Cases 9, 10) were inherited from phenotypically normal mothers. Eight confirmed anomalies were de novo in origin, and four families denied the parental studies. Out of ten TP deletions, seven had abnormal US findings (70%), while two of six confirmed duplications (33.3%). Almost all FPs (except Cases 39 and 48) had normal US findings. The confirmation rate was 29.4% per deletion (10/34), while 58.3% per duplication (7/12). None of the cases predicted for Prader-Willi or Angelman syndrome were confirmed (Appendix A). The size of the confirmed smallest duplication and deletion were 2.6 Mb and 5.6 Mb, respectively.

### 3.3. Screen Negative Cases Having Abnormal Ultrasound Findings

Two hundred fifty-six cases with screen-negative cfDNA test results for common aneuploidies were investigated due to abnormal ultrasound findings, and 28 (10.9%) resulted in chromosomal anomaly (Appendix A). These anomalies were four T21, one T18, and two T13 interpreted as false negative cfDNA testing. The remaining chromosome anomalies were not in the advertised coverage of the individual cfDNA test. We observed FGR and adverse pregnancy outcomes in both CPM cases (Case 8 with T7 and Case 9 with T16). One triploidy and one mosaic trisomy nine were detected in the other two cases (Case 10, Case 11). The remaining seventeen cases had structural anomalies; ten were detected by karyotyping and seven by microarray testing. Two anomalies were de novo balanced translocations. Almost all unbalanced SAs were de novo in origin except one (Case 16). Seven anomalies were diagnosed by molecular karyotyping, and the size of the CNVs varied between 660 Kb and 5.5 Mb.

## 4. Discussion

The expanded cfDNA testing allows the prediction of the risk for common trisomies as well as RCAs. If the test is positive for RCAs, genetic counseling is complicated since the knowledge about the incidence of an individual anomaly in early pregnancies is limited, the definitive exclusion of the mosaicism is challenging, and the phenotypic spectrum of mosaics is broad. Only a few reports shared the clinical and follow-up data with screen-positive cfDNA results for RCAs in the literature. This report aims to contribute to this field by presenting further data and achieving better diagnostic efficiency of cfDNA testing for RCAs.

The screening rate of RATs was reported in 0.1% to 1.1% of cases in cfDNA testing studies [31,32,33,34,35]. This study cannot give the screening rate, but RATs in our series accounted for 6.7% of screen-positives. Confirmatory testing in screen positives requires a decision between invasive procedure options; CVS or AC. Since non-mosaic RATs are usually lethal, the primary diagnostic challenge is the confirmation of mosaicism in fetal tissues (TFM) and excluding symptoms of mosaicism by expert sonography. AC is the most frequently applied technique in many centers [26]. Still, a significant disadvantage is the waiting period after a cfDNA test in the first trimester, increasing the probability of pregnancy termination with an unconfirmed cfDNA test result. Confirmatory testing with CVS has been a matter of considerable debate. Van Opstal and Srebniak [27] analyzed CVS data to create a modeling strategy. They recommended CVS for trisomies, mostly restricted to the cytotrophoblast (CPM-1; T3, T7, T8, T9, T20) and AC for other trisomies. They also recommended that UPD should be excluded if chromosomes with known imprinted regions (T6, T7, T11, T14, T15, T16, T20) are involved. Mardy and Wapner [21] favored CVS as a primary confirmatory test supplemented by AC only if required. The placental investigation can help elucidate the causes of false positive/negative results of cfDNA testing for aneuploidies. In this series, we were able to search both fetal and placental tissues in 13 cases and only chorionic villi in four cases. Predicted trisomies were confirmed as TP in 23.5% or as CPM in 52.9%, while 23.5% were not confirmed (FPs). If all 40 cases of our series (including the group of FP/CPM?) take into consideration, as expected, these rates decreased (TPs 17.5%, CPMs 22.5%, FPs 10%) (Table 1). In recent publications, PPV for RATs differs from 6% to 28.6% depending on the used tissue and techniques. Our TP rates (17.5% and 23.5%) were in agreement with these studies [36,37,38,39].

Cytogenetic testing of the term placenta may be more of academic interest [40,41]. However, we aimed to test term placental tissues in cases of suspected mosaicism if the parents consented to search for the efficiency of cfDNA testing. Mosaicism is an important biological phenomenon and can cause diagnostic pitfalls. Most of the mosaics (about 80%) observed in CVS are CPM, but up to 10% of apparent CPMs may be cryptic TFM [42]. If CPM-3 mosaicism is detected, the risk of being TFM is higher than for CPM-1 (40% vs. 3.7%) [20]. Uncultured cells might reflect a more realistic distribution of cell lines, and I-FISH on uncultured samples is a reliable technique to determine the types of mosaicism [43]. The positive screening result for a RAT allows us to search the mosaicism of predicted anomaly using the I-FISH technique using relevant probes. I-FISH studies changed the classification of mosaicism in a total of four cases, from CPM to TFM (Case 5, 6, 20) and from CPM-1 to CPM-3 (Case 2) in our series (Appendix A). Exceptional cases from the early CVS series remind us that diagnosing an abnormal cell line in the fetoplacental unit does not necessarily mean handicap [44]. If CVS results in both cytotrophoblasts and mesenchymal core are normal, the fetal tissues are most likely to be normal and still have a risk of 0.03% for a false negative result due to the test limitations for mosaicism [45].

The association between CPM and FGR or adverse pregnancy outcomes except for T16 is conflicting in the literature [16,22,23,24,25,46]. We observed FGR and adverse pregnancy outcomes of four CPM cases associated not only with T16 but also with T4, T7, and T22. In this series, we have not consciously encountered maternal malignancies or other conditions as possible causes of false positives [47].

In keeping with the literature findings, most were false positives for SAs in our series (overall PPV 32.7%). Among recurrent SAs, the highest PPV was found for del22q11.2 (33.3%), similar to the published studies (21% and 29%), likely due to its higher prevalence (1:2500) than other SAs [48,49]. PPV for 4p and 5p deletions seems to be also 33.3%. However, the number of cases is small to suggest. Although our confirmed cases were presented mostly with pathological ultrasound findings, the phenotype of the MMSs, including 22q11.2 deletion syndrome, is variable. Therefore, ultrasound examination may remain uninformative, especially in the first trimester of pregnancy. An early non-invasive screening test for MMSs has been helpful [29,50,51]. On the other side, cfDNA testing may detect the benign or unknown significance of CNVs, which would lead to further tests and parental anxiety, as in our cases with del(13)(q14.3q21.1) and del(14)(q21.1q21.2) (Cases 9–10, Appendix A). If the cfDNA result is positive for known benign CNVs, parental tests may be initially offered to prevent unnecessary invasive procedures and anxiety in the family. Another interesting observation was that two patients were referred with a predicted deletion on chromosome 10 with the same breakpoints (Cases 37 and 38). cfDNA tests were performed by different companies. Deletions in both cases were not confirmed. It is necessary to share similar observations and experiences to explain the causing factors in false positives.

We observed a higher confirmation rate for duplications than for deletions (PPV 58.3% vs. 29.4% per anomaly). There were similar results in the interstitial/terminal aberrations category of the TRIDENT2 study since 10 out of 12 duplications, and 17 out of the 43 deletions were confirmed in the mother or fetus [36]. To the best of our knowledge, the higher performance of the cfDNA testing for duplications was not reported in the literature. Two small deletions (1.25 Mb and 2 Mb) (Cases 8 and 11) were not detected in cfDNA testing, but they were out of the test coverage (Appendix A).

The debate on optimal coverage of cfDNA testing (common trisomies only vs. expanded testing) is ongoing. We diagnosed a significant number of unbalanced chromosome anomalies (28 in 256 cases, 10.9%) detected following a screen-negative cfDNA test result. Among them, 20 were out of the scope of the screening test. However, the detection of seven common trisomies (false negative cfDNA testing) underscores the importance of expert sonography to identify pregnancies at high risk for chromosome anomalies and the need to apply invasive testing in these situations. The findings of this study provide helpful information for laboratory and clinical management of RCAs predicted by expanded cfDNA testing. If the cfDNA test is positive for RATs, I-FISH with relevant probes should be included in the cytogenetic workflow. Investigation of chorionic villi tissue, even from the term placenta, would help to explain the FPs. In proven CPM cases with normal US findings, the families and physicians should be warned about the potential risk for poor pregnancy outcomes. Since the diagnostic accuracy of CVS and AC is equal for SAs, CVS seems to be the favored approach for rapid diagnosis.

The limitations of this study were (1) Screening tests were done by different commercial companies. Additionally, we don’t know how many cfDNA tests were done and do not give screening rates for rare chromosome anomalies. (2) We are not informed about the early ultrasound findings (at the time of the screening test). (3) As a reference center, most of the complicated cases are referred to our clinic. Therefore, the screen-positive cases for rare autosomal anomalies might have been selectively referred to us, which would change the percentage of rare anomalies in the whole series. (4) The term placenta studies could not be performed in about half of the cases with positive test results for rare autosomal aneuploidies. (5) All newborns could not be clinically examined due to the distance of the residential location of the families, and thus, we obtained the data by oral communication.

Dealing with the complexities of mosaicism in the fetoplacental unit and its diagnostic consequences still requires expertise in traditional cytogenetic techniques and current molecular cytogenetic test approaches. A prognostic assessment in these cases also relies on expert sonography, and increased experience will allow for achieving higher PPVs for RCAs.

## 5. Conclusions

The biology of early pregnancy and the concept of non-invasive screening are complex, and an adequate sharing of information with patients and service providers is an ongoing challenge.

We suggest that if cfDNA testing uncovers CPM, this does not mean cfDNA testing is “false positive.” The detection of CPM needs adequate laboratory work and specific genetic counseling, including backup risk for low-level true fetal mosaicism and a higher risk for poor pregnancy outcomes. Testing of cfDNA seems to be an efficient screening tool with a 23.5% FP rate in CVS for RATs. If AC is preferred to avoid the CPM, I-FISH studies using a specific probe for predicted trisomy should be included in the workflow.

If cfDNA testing predicts a known benign structural anomaly, the maternal genome can be evaluated before an invasive procedure.

The efficiency of cfDNA testing seems to be higher for duplications than deletions.

Even in cases with negative cfDNA testing, expert fetal ultrasonography is a reliable tool to identify high-risk pregnancies.

## Figures and Tables

**Table 1 genes-13-02389-t001:** The outcomes of the cases with a positive cfDNA test result for rare autosomal trisomies.

Predicted Rare Autosomal Aneuploidy	∑ *n*	True Positives *n*	CPM *n*	False Positives *n*	FP/CPM? * *n*
Trisomy 7	8	1	1	1	5
Trisomy 16	7	1	2	0	4
Trisomy 22	6	1	1	1	3
Trisomy 3	4	1	0	1	2
Trisomy 10	3	1	1	0	1
Trisomy 15	3	1	0	0	2
Trisomy 4	2	0	1	1	0
Trisomy 20	2	0	2	0	0
Trisomy 8	2	0	0	0	2
Trisomy 9	1	0	1	0	0
Trisomy 12	1	1	0	0	0
Monosomy 7	1	0	0	0	1
Total (%)	40	7 (17.5%)	9 (22.5%)	4 (10%)	20 (50%)

* FP/CPM? This group includes cases in which CPM could not be excluded because the placental tissue was not available.

**Table 2 genes-13-02389-t002:** The outcomes of the cases with a positive cfDNA test result for structural chromosome anomalies.

Structural Anomalies	∑ *n*	True Positives *n*	False Positives *n*	Positive Predictive Value
Deletions				
del (22)(q11.2)	9	3	6	33.3%
del (1)(p36)	3		3	
del (4)(p) *	3	1	2	33.3%
del (5)(p) *	3	1	2	33.3%
del(10)(q21.1q21.3)	2		2	
del (2q?) *	1		1	
del (2)(q24.3q31.2)	1		1	
del (3)(q11.2q13.13)	1	1		
del (7)(q21.11q22.1)	1	1		
del (7)(q21.13q31.32)	1		1	
del (10)(q25.2q26.3)	1		1	
del(10)(p subtel)	1		1	
del (13q?) *	1	1		
del (14)(q21.1)	1	1		
del (14)(q31.1q31.5)	1		1	
del (16)(q14.1q16.1)	1	1		
del (15)(q11.2q11.3)	1		1	
del (18)(q12.2q12.3)	1		1	
Total	33	10	23	30.3%
Duplications				
dup (22)(q11.2)	1	1		
dup (8)(q22.3q23.1)	1	1		
dup (4)(p16.3p15.2)	1	1		
dup (16p) *	1	1		
dup (1)(q25.3q32.1)	1		1	
dup (4)(q27q31.1)	1		1	
dup (8)(p11qter)	1		1	
dup(21)(q21.1q22.2)	1	1		
tetrasomy (12p)	1		1	
Total	9	5	4	55.6%
Double anomalies				
dup(3p)/dup(15q) *	1	1		
mosdup(7)(q33q36)/del (18)(q22q23)	1		1	
Y/2 abnormal CNVs	1		1	
Total	3	1	2	33.3%
Prader Willi/Angelman Syndrome	4		4	
SAs Total	49	16	33	32.7%

* declared by the physician.

## Data Availability

The authors have no further data for this study.

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
