# Peer review of "Clinical, Cytogenetic and Molecular Cytogenetic Outcomes of Cell-Free DNA Testing for Rare Chromosomal Anomalies"

_genes, 2022, doi:10.3390/genes13122389_

Round 1

Reviewer 1 Report

Your retrospective study focuses on the confirmatory and follow up data in pregnancies with positive NIPT results for rare chromosomal anomalies (RCAs).

There are some important points that, in my opinion, make the paper not suitable for publication:

·      The message : “The accuracy of detecting  RCAs is still relatively poor and should be improved” is not new (PMID: 30828085, 34895207, 36099005).

·      The small sample size with 887 cases does not allow to obtain statistically significant new data.

·      You compared your findings with previously published studies on the field and the comparison, which occupies most of the discussion, looks more like a detailed literature review than a discussion useful to highlight the novel data of your study.

Minor points:

The article is difficult to read

In Supplementary table the second sheed is about MMSs not about SCAs

Author Response

Response to Reviewer 1 Comments

We thank the Reviewers for their valuable comments. We believe that the manuscript has developed with their important contributions. Since the manuscript involved the results of a single center, the number of cases might be relatively small. However, we included more cases in the analysis and enhanced the Results and Discussion sections accordingly. The text is rewritten in several sections thus, we did not highlight the changes within the text.

Point 1: “The accuracy of detecting  RCAs is still relatively poor and should be improved” is not new (PMID: 30828085, 34895207, 36099005).

Response 1: We thank the Reviewer for their valuable comment. The proposed articles have been reviewed, and although the topic is not new, we think that it will contribute to the literature in terms of being a new series studied in a single center and offering a new perspective and approach especially to rare cases.

Point 2:  The small sample size with 887 cases does not allow to obtain statistically significant new data.

Response 2: We thank the Reviewer for their valuable comment. We included new cases to the series, and the number was increased to 912. Although the number in the case series seems to be small, we think that it will contribute to similar cases in terms of approach. The study involves many new cases that have not been mentioned in the literature before, rather than a statistically significant new result.

Point 3:  You compared your findings with previously published studies on the field and the comparison, which occupies most of the discussion, looks more like a detailed literature review than a discussion useful to highlight the novel data of your study.

Response 3: We thank the Reviewer for drawing attention to this issue.  We re-evaluated the overall manuscript from this perspective.

Minor points:

Point 4: The article is difficult to read

Response 4: We thank the Reviewer for their comment. We reviewed the text and reorganized the sentences to make them easier to follow.

Point 5: In Supplementary table the second sheed is about MMSs not about SCAs

Response 5: We thank the Reviewer for their comment. The second sheet of the Supplementary Table includes larger and complex RCAs as well as MMSs, we evaluated all the cases in this group in this category.

Reviewer 2 Report

Title: Clinical, Cytogenetic and Molecular Cytogenetic Outcomes of Cell-Free DNA Testing for Rare Chromosomal Anomalies

General Comments:

The work presented in the manuscript provide the comparative data on the efficacy of  Cell-Free DNA Testing for Rare Chromosomal Anomalies (RCA) in prenatal diagnostics. The authors gives a review of retrospective data analysis of 887 cases with positive and negative results with cfDNA for RCAs. Overall, the manuscript is not well organized to convey the objective of this study, namely, relationship between expanded tests and positive predictive values (PPV) for RCAs. This relationship depends on the prevalence of the rare fetal anomaly affects the PPV.

This is not made clear in various sections of the manuscript, i.e. abstract, introduction, methods, results, discussion and conclusion.

Specific Comments:

1.      Abstract:  structure needs to be revised. The authors should consider giving the results of chorionic villus sampling, amniocentesis, fetal blood sampling, and term placenta samples separately. This will make it more organized and clearer for the audience.

2.      Line 27-30 needs to be improved to state the goal of the study in a more clear and concise manner

Introduction

Line 53-80 to be more concise focusing on the cfDNA.

Line 63-70 can be deleted because it is well known and published. The authors can briefly described the Types 1-3 of mosaicism without going into details and frequencies

Line 87… Validated cfDNA…… Is this a goal of this study? If it is, then it should be stated clearly

Line 90-95 needs to be revised to give the aim of the study more clearly. The authors are recommended to give their aims as 1)…2)…. 3) ….and so on

Methods: Technical details of chromosome microarray need to be included. Threshold and log2 ratios for loss and gain need to be specified.

Results:

The data in tables should indicate all chromosome abnormalities according to ISCN2000 guidelines.

Discussion

Authors are recommended to organize the discussion by the sample type studied, namely, chorionic villus sampling, amniocentesis, fetal blood sampling, and term placenta. Authors should have a separate paragraph in the discussion for each type of sample. In the current format, the whole discussion appears poorly organized.

Conclusion:

Authors have presented the conclusion with general information without giving the specific achievements of their aims. It is recommended to revised it to give the specific conclusion from their study. They can also state their opinion for future studies to support the results of this study.

Author Response

Response to Reviewer 2 Comments

We thank the Reviewers for their valuable comments. We believe that the manuscript has developed with their important contributions. Since the manuscript involved the results of single center, the number of cases might be relatively small. However, we included more cases in the analysis and enhanced the Results and Discussion sections accordingly. The text is rewritten in several sections, thus we did not highlight the changes within the text.

General Comments:

Point 1: The work presented in the manuscript provide the comparative data on the efficacy of  Cell-Free DNA Testing for Rare Chromosomal Anomalies (RCA) in prenatal diagnostics. The authors gives a review of retrospective data analysis of 887 cases with positive and negative results with cfDNA for RCAs.

Overall, the manuscript is not well organized to convey the objective of this study, namely, relationship between expanded tests and positive predictive values (PPV) for RCAs. This relationship depends on the prevalence of the rare fetal anomaly affects the PPV.

This is not made clear in various sections of the manuscript, i.e. abstract, introduction, methods, results, discussion and conclusion.

Response 1: We thank the Reviewer for their valuable comment. We re-evaluated the overall manuscript in this perspective and reorganized the text to clarify the aim of the study and to interpret the results better.

Specific Comments:

Point 2: Abstract:  structure needs to be revised. The authors should consider giving the results of chorionic villus sampling, amniocentesis, fetal blood sampling, and term placenta samples separately. This will make it more organized and clearer for the audience.

Response 2: We thank the Reviewer for drawing attention to this issue.  Since, the findings in the Results section was organized according to RATs and SCAs and Discussion section was organized accordingly and the Abstract section was structured following the similar classification. The reason we preferred to structure the manuscript according to the type of indications not the invasive procedure is to make the Discussion section easy to follow. 

Point 3: Line 27-30 needs to be improved to state the goal of the study in a more clear and concise manner

Response 3: We thank the Reviewer for their valuable comment. We overviewed the Abstract section. Lines 27-30 was replaced by the following sentence.

“This study provides further data to assess the efficiency of expanded cfDNA testing for RATs and SCAs, cfDNA test efficiency seems to be higher for duplications than for deletions, evidence for the role of expert ultrasound in identifying pregnancies at increased risk for chromosome anomalies, even in pregnancies with screen-negatives. Furthermore, we discussed the efficiency of CVS vs. AC in screen-positives for RATs.”

Introduction

Point 4:  Line 53-80 to be more concise focusing on the cfDNA.

Response 4: We thank the Reviewer for their valuable comment. We overviewed the paragraph between the lines 53-80. Since the results are discussed in accordance to the CPM types we would to keep the main text of the paragraph. We believe that for the readers not closely familiar to CPM mechanisms the remaining of the manuscript will not be easy to follow if this section is omitted. 

Point 5 : Line 63-70 can be deleted because it is well known and published. The authors can briefly described the Types 1-3 of mosaicism without going into details and frequencies

Response 5: We thank the Reviewer for their valuable comment. We simplified and shortened the sentences between the Lines 63-70.

“Three different types of CPM are defined; CPM-1 (abnormality only in cytotrophoblasts); CPM-2 (abnormality only in the mesenchyme); and CPM-3 (abnormality in both cyto-trophoblasts and mesenchyme) [18]. Mosaicism is observed in about 1 to 2% of CVS. Of those, 72 to 87% are CPM. However, the likelihood of TFM still is about 13% in CPM cases [19,20]. CPM-2 and -3 are more likely to be associated with TFM, and therefore differential diagnosis between the subgroups of mosaicism is important. Interphase fluorescence in-situ hybridization technique (I-FISH) with relevant probes is helpful in searching the mosaicism [21].”

Point 6: Line 87… Validated cfDNA…… Is this a goal of this study? If it is, then it should be stated clearly

Response 6: We thank the Reviewer for drawing attention to this issue. We omitted the sentence on Line 87.

Point 7: Line 90-95 needs to be revised to give the aim of the study more clearly. The authors are recommended to give their aims as 1)…2)…. 3) ….and so on

Response 7: We thank the Reviewer for drawing attention to this issue. We overviewed this paragraph and replaced with the text below.

“Here, we present the confirmatory test results, fetal ultrasound findings, and clinical follow-up of the screen-positives for RCAs from the perspective of a tertiary care center. We discuss the efficiency of the different invasive procedures and laboratory techniques in screen-positives for RATs and SCAs. Furthermore, we share the cytogenetic findings of the screen-negative cases for common aneuploidies evaluated due to the ultrasound pathologies. Since the biological nature is complex and the frequency of the RCAs is unknown in early ongoing pregnancies, sharing the experiences with positive cfDNA testing for RCAs might be helpful to improve the diagnostic approaches and to give adequate pre- and posttest genetic counseling.”

Point 8: Methods: Technical details of chromosome microarray need to be included. Threshold and log2 ratios for loss and gain need to be specified.

Response 8: We thank the Reviewer for drawing attention to this issue. The microarray analysis criteria is written in detail in accordance with the recommendations.

Results:

Point 9: The data in tables should indicate all chromosome abnormalities according to ISCN2000 guidelines.

Response 9: We thank the Reviewer for their valuable comment. Chromosomal anomalies were given according to ISCN 2000. However, full cfDNA information of some cases could not be reached. In the Supplemental Table, we presented the anomalies as reported by the physician. Since the anomaly was not confirmed in these cases, the break points of the anomaly could not be given.

Discussion

Point 10: Authors are recommended to organize the discussion by the sample type studied, namely, chorionic villus sampling, amniocentesis, fetal blood sampling, and term placenta. Authors should have a separate paragraph in the discussion for each type of sample. In the current format, the whole discussion appears poorly organized.

Response 10: We thank the Reviewer for their valuable comment. Since, the findings in the Results section were organized according to RATs, and SCAs and Discussion section was organized accordingly, and the Abstract section was structured following a similar classification. However, we overviewed the manuscript thoroughly and restructured the text to make it clearer and easier to follow.

Conclusion:

Point 11: Authors have presented the conclusion with general information without giving the specific achievements of their aims. It is recommended to revise it to give a specific conclusion from their study. They can also state their opinion for future studies to support the results of this study.

Response 11: We thank the Reviewer for their valuable comment. We reorganized the conclusion and replaced it with the text below.

“The biology of early pregnancy and the concept of non-invasive screening are complex, and an adequate sharing of information with patients and service providers is an ongoing challenge.

We suggest that if cfDNA testing uncovers CPM, this does not mean cfDNA testing is “false positive.” The detection of CPM needs adequate laboratory work and specific genetic counseling, including backup risk for low-level true fetal mosaicism and a higher risk for poor pregnancy outcomes. cfDNA testing seems to be an efficient screening tool with a 23.5% FP rate in CVS for RATs. If AC is preferred to avoid the CPM, I-FISH studies using a specific probe for predicted trisomy should be included in the workflow.

If cfDNA testing predicts a known benign structural anomaly, the maternal genome can be evaluated before an invasive procedure.

The efficiency of cfDNA testing seems to be higher for duplications than deletions.

Even in cases with negative cfDNA testing, expert fetal ultrasonography is a reliable tool to identify high-risk pregnancies.”

Reviewer 3 Report

Paper by Basaran et al. is focused on the use of Cell-free DNA test to identify pregnancy at risk of rare chromosome anomalies. Data reported indicate that both invasive procedures of prenatal diagnosis and integrated ultrasound approach seem to be necessary because cfDNA  test applied to rare chromosome anomalies have a low rate of discrimination among false and true positives and in particular among true and false negatives. Data are well presented and discussion adequate.

On the other hand as well known the frequency of not hereditary chromosome anomalies increase with age and, in general, women >35 years old have an higher risk of pregnancies with chromosomal aberrations. I suggest to evaluate the effect of age on test performances. In other words would be of certain interest if PPV and NPV of cfDNA  test applied to rare Chr anomalies are dependent or not on the age of the mother.

Author Response

Response to Reviewer 3 Comments

We thank the Reviewers for their valuable comments. We believe that the manuscript has developed with their important contributions. Since the manuscript involved the results of single center, the number of cases might be relatively small. However, we included more cases in the analysis and enhanced the Results and Discussion sections accordingly. The text is rewritten in several sections, thus we did not highlight the changes within the text.

Paper by Basaran et al. is focused on the use of Cell-free DNA test to identify pregnancy at risk of rare chromosome anomalies. Data reported indicate that both invasive procedures of prenatal diagnosis and integrated ultrasound approach seem to be necessary because cfDNA  test applied to rare chromosome anomalies have a low rate of discrimination among false and true positives and in particular among true and false negatives. Data are well presented and discussion adequate.

Point 1: On the other hand as well known the frequency of not hereditary chromosome anomalies increase with age and, in general, women >35 years old have an higher risk of pregnancies with chromosomal aberrations. I suggest to evaluate the effect of age on test performances. In other words would be of certain interest if PPV and NPV of cfDNA  test applied to rare Chr anomalies are dependent or not on the age of the mother.

Response 1: We thank the Reviewer for their kind evaluation. In order to evaluate the influence of maternal age on cfDNA results in RAT and SCA groups, we performed One-way ANOVA. However, the results were non-significant. Additionally, the mean age and the ranges of maternal age between the groups were not comparable between the groups. Thus, we did not include these results in the manuscript.

The results of the statistical analysis are given in the included file.

Round 2

Reviewer 1 Report

I have read the new version of the paper and now, in my opinion,  it can be published.

Author Response

We thank the reviewer for their valuable comments.